# Modulatory Mechanisms of Pathogenicity in *Porphyromonas gingivalis* and Other Periodontal Pathobionts

**DOI:** 10.3390/microorganisms11010015

**Published:** 2022-12-21

**Authors:** Sara Sharaf, Karolin Hijazi

**Affiliations:** Institute of Dentistry, University of Aberdeen, Aberdeen AB25 2ZR, UK

**Keywords:** *Porphyromonas gingivalis*, molecular pathogenesis, virulence

## Abstract

The pathogenesis of periodontitis depends on a sustained feedback loop where bacterial virulence factors and immune responses both contribute to inflammation and tissue degradation. Periodontitis is a multifactorial disease that is associated with a pathogenic shift in the oral microbiome. Within this shift, low-abundance Gram-negative anaerobic pathobionts transition from harmless colonisers of the subgingival environment to a virulent state that drives evasion and subversion of innate and adaptive immune responses. This, in turn, drives the progression of inflammatory disease and the destruction of tooth-supporting structures. From an evolutionary perspective, bacteria have developed this phenotypic plasticity in order to respond and adapt to environmental stimuli or external stressors. This review summarises the available knowledge of genetic, transcriptional, and post-translational mechanisms which mediate the commensal-pathogen transition of periodontal bacteria. The review will focus primarily on *Porphyromonas gingivalis*.

## 1. Introduction

Periodontitis is a highly prevalent inflammatory condition of the tooth-supporting structures associated with bacteria of the “red complex”, which comprise *Porphyromonas gingivalis*, *Treponema denticola* and *Tannerella forsythia*. These bacterial species interact amongst themselves and with other species as plaque forms in the subgingival sulcus [1]. Under specific environmental conditions, periodontal bacteria produce proteases, most notoriously gingipains and dentilisin, that drive protein degradation and ultimately contribute to tissue destruction of the periodontium [2]. However, bacterial protease production is only one of a plethora of virulence mechanisms associated with periodontal bacteria, many of which are very well studied for their ability to subvert the immune system and cause local and systemic inflammation. *P. gingivalis* is considered a keystone bacterium in the pathogenesis of periodontitis by driving an imbalance between symbionts and pathobionts of the oral microbiota (dysbiosis) [3,4] and interfering with the complement cascade amongst other immune responses [5]. This review presents published information regarding the mechanisms by which periodontal pathobionts, in particular *P*. *gingivalis*, are able to transition from harmless colonisers of the subgingival environment to virulent pathogens. 

From an evolutionary perspective, bacteria have developed a phenotypic plasticity that allows survival responses and adaptation to environmental stimuli or external stressors. This pathogenic shift may be viewed in the context of dysbiosis, a term used to broadly describe imbalances of the microbiota which cause disease [6], and therefore is likely relevant to the pathogenesis of periodontitis [7]. Dysbiosis may be induced by environmental changes such as physical disruption of the epithelial barrier, antimicrobial treatments, and immune deficiencies. At the most basic level, oral microbiota and dysbiosis studies have focused on the loss of microbial diversity, that is the increase in abundance of ‘pathobionts’ and the depletion of ’symbionts’ [8,9]. The term pathobiont is used to describe species which are present as commensal microbes but under certain environmental conditions can induce or advance disease [10]. Against a backdrop of metagenomic studies that supports periodontitis as a dysbiotic disease, the specific mechanisms by which periodontal bacteria shift from a commensal to a pathogenic state are not fully understood. 

External stressors can induce bacterial phenotypic changes for their adaptation and survival. Phenotypic change may also result from the accumulation of non-synonymous single-nucleotide mutations. Such mutations are spontaneous and, at least in part, driven by aberrant DNA repair mechanisms [11]. However, variants of *P. gingivalis* clones generated in vitro by serial sub-culturing are characterised by loss of virulence (loss of pigmentation in blood agar growth, decreased proteolytic activity, depleted haemagglutination and higher susceptibility to complement killing) suggesting that spontaneous mutations are less likely contributors to pathogenicity [12].

Environmental factors are the major drivers of phenotypic change in most bacteria including periodontal species. For example, temperature is a well-known regulator of the activity of transcriptional regulators, kinases, and chaperones [13,14]. In the case of *P. gingivalis*, temperature elevation has been shown to alter the structure of Lipid A, a major component of lipopolysaccharide (LPS). *P. gingivalis* grown at temperatures representative of the inflammatory environment shows reduced diversity of Lipid A structural forms, namely the predominance of monophosphorylated and diphosphorylated penta-acylated forms which are more potent Toll-like receptor 4 (TLR4) agonists [15]. Similarly, haemin concentration in in vitro growth culture can modulate TLR4/2 activation and E-selectin expression by *P. gingivalis* through their Lipid A structure. Under elevated haemin concentration, *P. gingivalis* showed ‘immunosuppressive’ properties (downregulation of TLR signalling, reduced levels of TNF-α and elevated interleukin-10 production) [16,17,18]. Under haemin-limiting conditions, *P. gingivalis* showed ‘immune-evasive’ properties, for example, increased resistance to host cationic antimicrobial peptides and lower LPS-mediated neutrophil-priming capacity, as well as increased gingipain activity, extracellular vesicles release and nutrient storage [19]. Extracellular pH also plays a major role in driving survival and dominance of periodontal pathobionts. While *T. denticola* and *T. forsythia* show increased survival at acidic pH, *P. gingivalis* predominates at alkaline pH [20,21]. 

There is evidence that *P. gingivalis* can activate manganese transport in addition to iron transport as a compensatory mechanism to enhance survival in low iron environments. This mechanism is mediated by *feoB1* and *feoB2* encoding iron and manganese ion transporters, respectively, which also play a role in protection from oxidative stress [22]. Superoxide dismutases (*sod*) of *P. gingivalis* can utilise iron or manganese as a cofactor to catalyse a reaction that converts the superoxide radicals into hydrogen peroxide and molecular oxygen [23]. This is thought to be part of the evolutionary adaptation aiding survival of *P. gingivalis* in iron-depleted habitats.

The production of Reactive Oxygen Species (ROS) increases as a result of oxidative stress produced by host defences or release by early bacterial colonisers of the oral cavity, for example *Streptococcus ssp*. Biomarkers of oxidative stress include protein carbonylation, 8-hydroxy-2-deoxyguanosine and lipid peroxidation, which are products of oxidation produced as a result of damage to proteins, DNA and lipids, respectively [24]. Elevated levels of ROS released by neutrophils during infection induce a cytotoxic effect on both human gingival fibroblasts and periodontal bacteria. Moreover, they are secondary messengers in the regulation of cell signalling, cellular homeostasis, and cell apoptosis [25,26]. ROS are sensed by transcriptional regulators which activate a range of survival mechanisms. *Fusobacterium nucleatum* is considered a scaffold bacterium linking early colonisers to late colonisers of the biofilm and is essential for the continued survival of other anaerobic bacteria in the presence of ROS [27,28]. *F. nucleatum* uses the alkyl hydroperoxide reductase (AhpC) system for detoxification releasing less ATP, and increasing chaperone proteins to counteract oxidative stress [29]. *P. gingivalis* expresses similar strategies through the AhpC antioxidation system to provide cellular resistance in the earlier stages of oxidative stress. Furthermore, it can express chaperone proteins in response to oxidative stress to refold misfolded proteins [29,30].

Periodontal bacteria are capable of utilising a vast complement of mechanisms to respond to enviromental stresses and also evade/subvert host responses. In this review, we summarise the literature regarding the genetic, transcriptional, and post-translational mechanisms which underpin the phenotypic plasticity of periodontal pathobionts, with primary focus on *P. gingivalis.*
Table 1 and Figure 1 provide an overview of all the mechanisms discussed in this review.

## 2. Genetic Mechanisms of Pathogenicity in Periodontal Bacteria

Bacterial genetic diversity enhances bacterial fitness and survival in adverse environmental conditions. This often results in enhanced bacterial pathogenicity, primarily through the ability to overcome and subvert human innate and adaptive immune responses. Genetic diversity is mediated partly by mutations and gene polymorphisms, and partly by horizontal gene transfer through conjugation or transduction. In addition, extracellular DNA may also be the source of antimicrobial resistance gene acquisition [101]. DNA transfer through outer membrane vesicles (OMVs) has also been described in periodontal bacteria [61]. 

### 2.1. Virulence Gene Polymorphisms and Genetic Diversity

The polymorphisms of several virulence genes are associated with phenotypic variation and increased pathogenicity of *P. gingivalis*. However, the evidence in this area is at best conflicting with many studies disputing a consistent association between gene polymorphisms and clinical disease, or even with in vitro pathogenicity. Here we briefly summarise key literature relating to the genetic diversity of genes encoding the major virulence factors of *P. gingivalis*.

Allelic diversity of key virulence genes of *P. gingivalis* is thought to be partly mediated by the acquisition of extracellular DNA through a transformation-like mechanism. In this context, clonal diversity is enriched, notwithstanding that the fittest strains survive and dominate within the community [62].

*P. gingivalis* possesses major and minor fimbriae which play an essential role in adherence, invasion of host cells, biofilm formation and immune evasion [102]. Fimbriae enhance the intra-cellular survival of *P. gingivalis* in macrophages through the hijacking of the TLR2 signalling pathway and internalisation by CD11b/CD18 [31]. The *fimA* gene, encoding the major fimbrial protein FimA, is the best studied in terms of allelic variations. The *fimA* gene exists as a single copy on the *P. gingivalis* chromosome and is considered the basis for the definition of six genotypes of *P. gingivalis*. However, there are conflicting reports on the phenotypic significance and impact on the virulence of *fimA* variants. For example, one study that recruited 115 patients suffering from chronic periodontitis and 136 healthy controls reported the predominance of *fimA* type II and IV genotypes in subgingival plaque samples from deep periodontal pockets and types I and III in plaque from healthy periodontium. Additionally, the periodontal sites of *fimA* type II predominance displayed higher retrieval frequency of *A. actinomycetemcomitans* and *T. forsythia* [32]. Another study showed that type II *fimA* is associated with enhanced adherence and invasion of epithelial cells compared to type I *fimA* [33]. Another study followed to demonstrate that *fimA* type III and IV exhibit improved invasion of host gingival cells [34]. The results of these studies indicate that *fimA* type II is frequently associated with a diseased periodontium. On the other hand, there are conflicting reports regarding the effect of each *fimA* genotype on the pathogenicity of *P. gingivalis* [33,34,35].

The *rag* locus found in some strains of *P. gingivalis* encodes two proteins: RagA, a TonB-dependent receptor which controls substrate-specific transport through the outer membrane, and RagB, a lipoprotein that constitutes an immunodominant outer membrane antigen. There are four allele variants of the *rag* locus: *rag-1* was first discovered in strain W50, but *rag-2*, *rag-3* and *rag-4* were subsequently characterised [36]. These alleles showed no evidence for a specific geographical predilection. In a collection of 168 isolates of *P. gingivalis* mainly derived from patients with periodontitis across 15 countries, *rag-2* was the predominant allelic form, while *rag-4* was the least [37]. In mouse models, inoculation with *rag-1* positive isolates showed an increased level of virulence and soft tissue destruction compared to other *rag* variants [36]. On the other hand, a study that investigated *rag* locus variants in subgingival plaque samples from patients with chronic periodontitis showed that *rag-1* and *rag-3* were the most prevalent allelic forms [38]. Another study that sought to investigate the genotypes present in cases of orthodontic gingivitis and mild/moderate periodontitis reported the predominance of *rag-3* and *rag-4* in these conditions [39]. 

Six unique capsular serotypes of *P. gingivalis* are described based on the composition of the K-antigen capsule, each displaying different pathogenic potential, immune-evasive and pro-inflammatory properties [44]. *P. gingivalis* strain W83 is a virulent serotype K1 encapsulated strain, while *P. gingivalis* ATCC 33277 lacks a capsule and is considered avirulent [45,46]. Encapsulated *P. gingivalis* W50 was more resistant to phagocytosis by murine dendritic cells and macrophages compared to its isogenic nonencapsulated mutant (PgC). Additionally, reduced expression of CLCN2, CRP, TGF-α, CXCR4, IL-17, and AGT was observed 1 hr post infection with the encapsulated strain compared to nonencapsulated PgC [47].

### 2.2. Horizontal Gene Transfer and Genetic Rearrangements

Acquisition of new phenotypes by periodontal bacteria can be mediated by horizontal gene transfer [103,104]. Horizontal gene transfer plays a significant role in genetic diversity and evolution of *P. gingivalis*, and by extension in the enhancement of its survival and virulence. At the simplest level, key virulence factors and antimicrobial resistance genes of *P. gingivalis* are thought to be the result of intra-species and inter-species DNA acquisition. 

Genetic rearrangements in *P. gingivalis* through mobile genetic elements are well documented. Conjunctive transposons and insertion sequences are likely incorporated in response to environmental stressors to enhance adaptability. The typical presence of antimicrobial resistance genes on conjugative transposons has also been implicated as a driver of the spread and increase of the antimicrobial resistance burden. CTnPg1 is a conjunctive transposon, the transfer of which is reported between *P. gingivalis* strains [48] and also between *P. gingivalis* and other periodontal bacteria, specifically *Bacteroides thetaiotaomicron* and *Prevotella oralis* [49]. The excision of CTnPg1 from the donor chromosome is mediated by the PGN_0094 integrase encoded within the transposon itself. CTnPg1 requires a 13-base pair sequence (TTTTCNNNNAAAA) to identify its insertion site and be incorporated within the recipient genome. Given the high variation of the target sequence across strains, the transposon may integrate across multiple sites within the *P. gingivalis* chromosome [49]. In addition, the recipient strain requires active RecA which is a recombinase responsible for DNA repair and maintaining genome integrity [50] to successfully receive CTnPg1. Importantly, CTnPg1 is likely a vehicle for the transfer of antimicrobial resistance as it includes genes encoding a sodium-mediated multidrug efflux pump, ABC transporters and HAE3 family efflux transporters which can be integrated into multiple sites on the recipient chromosome [49]. *P. gingivalis* also contains multiple insertion sequences (IS) which undergo transposition through shuffling from one genomic site to another, or retention of the original IS while producing a copy at a different site within the genome. Insertion sequences in *P. gingivalis* acquired several designations across studies, but they are mostly classified into ISPg1-ISPg7 [51,52]. Despite the frequency of IS shuffling, the representation of different ISPg varies greatly across strains of *P. gingivalis* [52]. Of note, is the identification of ISPg4 in the genome of more virulent strains of *P. gingivalis,* implicating ISPg4 (alias IS1598) as a unique marker of virulence [53]. However, a study later identified several duplicates of the virulent strain FDC 381 that lack the presence of ISPg4/IS1598, proving that *P. gingivalis* virulence is not significantly diminished by the absence of IS1598 [54]. Furthermore, miniature inverted-repeat transposable elements (MITEs) were discovered in high frequency in several *P. gingivalis* strains [105]. MITEs are short non-coding genetic elements that possess terminal inverted repeats and generate target site duplicates, they are usually AT-rich and preferentially insert into AT-rich regions. The role of MITEs in the modulation of gene expression is well documented [106]. A virulence-associated MITE termed E622 was identified in *Pseudomonas syringae* and implicated in the mobilisation of antimicrobial resistance genes through an antibiotic coupling mobility assay [107]. Another study identified MITE*_Aba12_* in *Acinetobacter baumannii* which confers heavy-metal resistance through mobilisation of resistance genes such as *mer* (mercury resistance gene) [108]. It is reasonable to expect MITEs within the *P. gingivalis* genome to influence gene expression in a similar mechanism. However, additional research is required to understand and confirm the full effect of these elements.

Gingipains are trypsin-like cystine proteases that contribute to most of the overall proteolytic activity of *P. gingivalis* during periodontal inflammation. Broadly, gingipains may be either arginine-specific or lysine-specific depending on their site of cleavage. Arginine-specific gingipains are encoded by *rgpA* and *rgpB* while lysine-specific gingipains are encoded by a single *kpg* gene. In addition to the proteolytic activity, arginine-specific gingipains are important mediators of pro-inflammatory responses where they directly induce NF-κB dependent production of hepatocyte growth factor (HGF) [109]. The enrichment of gingipain-encoding genes in OMVs suggests the involvement of OMVs as delivery vehicles for horizontal transfer of key virulence factors between different strains of *P. gingivalis* [110]. 

As mentioned above, the *rag* locus is a pathogenicity island associated with *P. gingivalis* virulence [40]. It is suggested that the locus is acquired from other bacteria within the microbial community such as *Bacteroides* species through horizontal gene transfer [111]. ISPg1 (IS1126) is present upstream of the *rag* locus while being flanked by 12-base pair inverted repeats; this insertion sequence consists of a single open reading frame and can generate a 5-base pair target site duplicate [111]. Multiple studies corroborated the *rag* locus as a horizontally transferred pathogenicity island that plays an important role in the virulence of *P. gingivalis* [40,111]. Subcutaneous inoculation of mice with *rag*-knock-out mutants resulted in reduced mortality compared to the *P. gingivalis* W50 wild-type inoculum [36]. Another study showed reduced spleen invasion by Δ*ragB P. gingivalis* W83 after subcutaneous inoculation compared to *P. gingivalis* W83 wild-type [41]. Decreased endothelial cell invasion was also shown for *P. gingivalis* Δ*rag* mutants compared to the parent wild-type as measured by antibiotic protection assays [42]. However, in a more recent study, despite the prevalence of *rag*-positive isolates in a periodontitis cohort over healthy controls, there was no difference in the ability to invade primary human fibroblasts by *rag*-positive strains over *rag*-negative isolates [43].

Clustered Regularly Interspaced Short Palindromic Repeats (CRISPR)-associated genes are found in *P. gingivalis*. As in other bacteria, CRISPR represents a critical protective mechanism from invasion by bacteriophages [55]. The genome editing activity of CRISPR-Cas systems occurs in three stages: (i) spacer acquisition through recognition of the spacer sequence and its integration into the CRISPR array; (ii) production of CRISPR RNA through cleavage of pre-CRISPR RNA; (iii) interference through mature CRISPR RNA recognition of foreign DNA [56,57]. CRISPR spacers are direct repeats separated by short DNA stretches, which act as a memory pool of past events and have possible targets within the *P. gingivalis* genome. These potential targets include coding sequences of multiple functions and sequences related to exogenous elements, such as bacteriophages and conjugative transposons [58]. Four CRISPR arrays were identified in *P. gingivalis* W83, namely CRISPR 30, CRISPR 36.1, CRISPR 36.2 and CRISPR 37 [59]. CRISPR arrays are responsible for regulating mobile genetic element acquisition in *P. gingivalis*, and a recent report suggested an important role for *cas3* in the modulation of potentially pathogenic phenotypes. *Cas3* is considered as the signature protein of type I CRISPR-Cas systems. *Cas3*-deletion mutants of *P. gingivalis* co-cultured with the monocytic cell line THP-1 showed significantly higher expression of *rgpA* and several genes associated with response to oxidative stress and iron uptake in *P. gingivalis*. On the other hand, genes encoding Tra proteins involved in pili formation and ribosomal proteins of both the large and small subunits were downregulated [60]. *Cas3*-deletion mutants also displayed a modest enhancement of pro-inflammatory properties (IL-1β, IL-6, and IL-10), suggesting that *cas3* plays a role in immune homeostasis [60]. Collectively, these findings suggest an overall pathogenicity downregulation by *cas3,* notwithstanding that the molecular mechanisms underpinning these effects remain unknown.

In addition to the well-characterised mechanisms of horizontal gene transfer (conjugation, transformation or transduction), vesicle-mediated DNA transfer has also been described in *P. gingivalis* strains [61,62]. OMVs are usually produced by budding from the cell surface of bacteria and enclose periplasmic content. As in other Gram-negative bacteria, *P. gingivalis* vesicles are 50–250 nm in diameter and have a protein profile that is similar to the outer membrane [63]. Indeed, proteins of the outer membrane of *P. gingivalis* that have been discovered in OMVs include PorV, LptO, RgpA and Kgp, all major contributors to the pathogenicity of *P. gingivalis* [64]. As a result, OMVs inherited the functions of these outer membrane proteins and exhibit features such as coaggregation with other oral bacteria, invasion of host cells and pro-inflammatory potential [65]. The molecular mechanism underpinning the formation and release of *P. gingivalis* vesicles remains poorly defined, but different mechanisms of OMV biogenesis were proposed [66,67]. In the biogenesis of OMV outer membrane proteins and lipoproteins are involved in peptidoglycan cross-linking and anchorage, specifically Lpp, NlpI, OmpA (Pgm6/7) [68]. Tol-Pal members are thought to stabilise the cellular envelope by linking the peptidoglycan layer with the inner membrane in Gram-negative bacteria [68]. It follows that any mutations or deletions of genes encoding Tol-Pal proteins can be reasonably expected to lead to the formation and release of OMVs. The O polysaccharide of LPS has also been implicated in the formation and protein packing of OMVs, mainly mediated by LPS-protein ionic interactions. Indeed, LPS modifications are known to upregulate OMV formation and release in other bacteria such as *Salmonella typhimurium* [69]. DNA packaging into outer vesicles may occur via uptake of extracellular DNA by free vesicles or DNA uptake into vesicles during cell death. However, neither of these mechanisms has been confirmed. Virulent factors and antibiotic resistance genes packaged into OMVs may play an important role in enhancement of bacterial survival and virulence mediated by horizontal gene transfer [70]. Important *P. gingivalis* genes such *fimA*, superoxide dismutase (*sod*), *hmuY* and *rprY* were over-represented in OMVs, implying that virulence factor DNA may be preferentially packaged by *P. gingivalis* into vesicles [61,62]. 

## 3. Transcriptional Mechanisms of Pathogenicity in Periodontal Bacteria

*P. gingivalis* induces the release of haem in the extracellular space through proteolytic activity such as that mediated by gingipains. Free haem is then sequestered by a haemophore-like protein (HmuY) through which it is delivered to the TonB-dependent outer-membrane receptor responsible for haemoglobin binding (HmuR) [71]. Ferric uptake regulator homologue (PgFur) binds to the *hmu* operon promoter and triggers the expression of *hmuR* and *hmuY* [72]. The expression of *pgfur* depends on the growth phase and iron/haem gradient in the growth environment. To better understand the role of PgFur in the virulence of *P. gingivalis*, a study compared the effect of *pgfur* inactivation in the avirulent strain ATCC 33277 versus the virulent strain A7436. Under high haem conditions the *pgfur* A7436 mutant strain displayed higher cell density during planktonic growth compared to the wild-type, while wild-type and the *pgfur* mutant strain of ATCC 33277 showed similar growth rates. Additionally, mutant strains of both virulent and avirulent *P. gingivalis* were highly sensitive to oxidative stress compared to the respective wild-types [73]. Moreover, the deactivation of the *pgfur* gene results in decreased adherence, reduced invasion of host cells and reduced intracellular survival [74]. HumY is resistant to several proteases including both gingipain and host proteases such as neutrophil elastase. This property contributes to bacterial persistence in adverse host environments and is reflected in the elevated levels of anti-HmuY antibodies observed in patients suffering from chronic periodontitis [75].

PgRsp is a member of the Crp/Fnr superfamily transcription regulators responsible for haem-binding and redox state sensing through haem-catalysed oxidation [76,77]. In support of these roles, *pgrsp* mutants showed a reduced uptake of haem mediated by HmuY in biofilm-forming conditions [76,78]. Further, PgRsp has been implicated in modulation of bacterial virulence. For example, upregulation of the expression of *kgp* and downregulation of the expression of *rgpB* and *rgpA* were observed when comparing the *pgrsp* knock-out mutant to the wild-type [76]. Inactivation of the PgRsp-encoding gene in *P. gingivalis* resulted in increased coaggregation with both *Prevotella intermedia* and *T. forsythia* whilst coaggregation with *Streptococcus gordonii* was not affected by mutagenesis [76]. The inactivation mutant displayed enhanced biofilm formation but diminished *fimA* expression and reduced survival in macrophages compared to the wild-type, suggesting an important role for *pgrsp* in regulation of pathogenicity [76]. 

Environmental stresses such as oxidative and nitrate stresses trigger transcriptional mechanisms of bacterial adaptation and survival which, in turn, may be deemed important for pathogenicity. For example, the transcriptional regulator RprY of *P. gingivalis*, induced as a response to oxidative stress, binds to the target promoter and represses the toxic effect of ROS by regulating the activity of the sodium-dependent ubiquinone oxidoreductase system. This system utilises the sodium gradient to produce energy for the cytochrome d oxidase (cydAB) operon which converts oxidative radicals into water. In aerobic conditions, *rprY* inactivation is lethal, presumably due to the decreased activity of the sodium-dependent ubiquinone oxidoreductase system regulated by RprY [79].

Extracytoplasmic function (ECF) sigma factors comprise a large group of transcriptional factors that regulate gene expression in response to environmental stresses. Six ECF sigma factors were described in *P. gingivalis* ATCC 33277: PGN_0274, PGN_0319, PGN_0450, PGN_0970, PGN_1108, and PGN_1740 [80], while in *P. gingivalis* W83 ECFs identified are PG0162, PG0214, PG0985, PG1318, PG1660, and PG1827 [81]. The *P. gingivalis* W83 isogenic single inactivation mutants of PG0985, PG1660 and PG1827 displayed reduced growth and survival upon exposure to hydrogen peroxide when compared to *P. gingivalis* W83 wild-type. Moreover, Rgp gingipain activity was reduced by 50% in the PG0162 knock-out and by 60% in the PG1660 knock-out mutant, while Kgp activity was reduced by 20% and 50% in the PG0162 and PG1660 mutants, respectively [81]. Synergy can occur between ECF sigma factors as observed for PG0162 and PG1660, jointly contributing to overall bacterial resistance to oxidative stress. However, the mechanism by which oxidative stress is sensed in this scenario and how ECF sigma factors are activated requires further study [81,82]. The PorX/PorY two-component system controls the expression of *por* genes encoding the type IX secretion system and ECF sigma factor SigP. The PorX/PorY system upregulates the transcription of the *sigP* gene, which in turn mediates the transcriptional activation of the *por* genes by binding to their promoters [83]. The *sigP*-knock-out of *P. gingivalis* ATCC 33277 showed reduced auto-aggregation compared to the respective wild-type, while *sigP* and *sigH* knock-outs both exhibited a drop in haemagglutination activity. When testing the effect of *sigP* on *P. gingivalis* virulence in a mouse infection model, the *sigP* mutant exhibited lower mortality rates compared to the W83 parent strain [80]. Another study confirmed this finding and showed a similar effect as a result of deletion of *porX* [84]. Additionally, the two-component system in *P. gingivalis* modulates the transcription of genes associated with biofilm formation, LPS modification and host cell invasion [83].

## 4. Post-Translational Mechanisms of Pathogenicity in Periodontal Bacteria

Post-translational modification of proteins mediates important structural and functional properties. A variety of protein post-translational changes can be induced by environmental changes and affect the cellular mechanisms that enhance bacterial survival and pathogenicity. Post-translational mechanisms deemed particularly important for pathogenicity of *P.gingivalis* include glycosylation and acetylation [112].

### 4.1. Acetylation

Lysine acetylation is achieved by lysine acetyltransferases and acetyl-coenzyme A. It generally mediates changes in the secondary structure of a protein and resulting binding properties. The presence of lysine acetylated enzymes in *P. gingivalis* was linked to carbohydrate, amino acid and lipid metabolism but not glycan biosynthesis [85]. These metabolic pathways are essential for the continued growth and persistence of *P. gingivalis* in the microbial community as they produce ATP and promote the release of short fatty acids required for bacterial growth and survival [85]. Protein acetylation leads to the activation of proteases which aids pathogenesis in periodontitis through the degradation of host extracellular matrix components, degradation of antimicrobial peptides and cytokines [86]. Proteins such as Mfa1 fimbrilin, Haemagglutinin protein HagA and Methionine gamma-lyase related to *P. gingivalis* pigmentation were identified as targets of acetylation. Acetylation of the Ferritin and Putative universal stress protein (UspA) has also been associated with *P. gingivalis* oxidative stress resistance [87]. Lysine acetylation plays a major role in the enhancement of bacterial survival and virulence as all three types of gingipains are acetylated [88]. 

RprY mediates oxidative stress response in *P. gingivalis*. RprY plays an essential role in the regulation of *P. gingivalis* virulence through type IX secretion system, which releases gingipains and peptidyl-arginine deiminase (PPAD), both considered major mediators of periodontal tissue destruction. Moreover, the reduction of RprY decreases Mfa1 fimbriae-mediated adhesion as observed by the retarded transcription of *mfa1* in ∆*rprY* mutants when compared to the wild-type [89]. RprY is acetylated in vivo in conjunction with its co-transcriptional partner downstream of RprY which is known as protein acetyltransferase (Pat) [90].

There is extensive overlap between lysine acetylation and lysine succinylation in *P. gingivalis* ribosomal proteins. It was observed that ten proteins responsible for glutamate and aspartate catabolism in *P. gingivalis* experience both acetylation and succinylation [87]. This overlap is consistent with findings in other bacteria, but its significance in the regulation of survival and virulence has not been well explored.

### 4.2. Glycosylation

Glycosyltransferases break down monosaccharides or oligosaccharides from an activated sugar donor (UDP-sugar) to different substrates, including carbohydrates, proteins and glycoproteins. *P. gingivalis* glycosyltransferases play a role in the synthesis of A-LPS and O-LPS lipopolysaccharide [91]. A-LPS anchors virulence proteins such as gingipain RgpB and Haemin-binding protein 35 (HBP35). The latter binds thioredoxin and haemin supporting bacterial haem acquisition especially in iron-depleted environments [92], hence affecting pathogenicity [91,92,93,94,95,96]. Gingipains themselves are also substrates of glycosyltransferases. Glycosylation of the arginine-specific proteinase and adhesin (RgpA) protects the enzyme against proteolytic degradation in adverse conditions induced by host responses at inflammed periodontal sites [97]. Virulence-modulating gene F (*vimF*) is a putative group 1 glycosyltransferase which regulates virulence and haem acquisition. Indeed, *vimF* inactivation results in reduced proteolytic activity and lower haemagglutination in *P. gingivalis* W83 cultured in vitro [93]. This is attributed to a glycosylation defect of haemagglutinin HagA and retarded gingipain maturation [93].

The glycosylation of surface proteins is critical to the regulation of host-microbial interactions. OMP85 is a highly conserved outer membrane protein of *P. gingivalis*, the glycosylation of which is mediated by the *galE* gene [96]. Investigations of the functional role of *galE* in *P. gingivalis* revealed a critical role in the modulation of biofilm formation. In a study where *P. gingivalis* ATCC 33277 and the isogenic *galE* knock-out mutant were incubated with an antibody targeting the outer-loop peptide of OMP85, it was observed that the antibody had an enhanced inhibitory effect on bacterial attachment and biofilm formation in the *galE* mutant over the wild-type. This suggested an imporant role of *galE*-mediated glycosylation in the maturation of adhesins and biofilm-forming factors [96]. 

O-glycosylation systems in *T. forsythia* mediate glycosylation of cell surface S-layer glycoproteins (TfsA and TfsB), which play an important role in bacterial virulence through promoting biofilm formation and bacterial aggregation [98]. A study measured the function of human monocyte-derived dendritic cells stimulated with three single gene knock-outs of glycosyltransferases *gtfS, gtfI and gtfE* in comparison with the respective wild-type. Here, *T. forsythia gtfE* knock-outs induced marked elevation of IL-1β, IL-12, IL-10 and IL-23, suggesting a role for *gtfE* in Th17 activation. On the other hand, these cytokines were significantly reduced in *gftS* and *gftI* single mutants compared to the wild-type suggesting Th17 suppression. These findings suggest a complex immunomodulatory role for *T. forsythia* [99].

Flagellar proteins (FlaA, FlaB and FlaB1) of *T. denticola* are heavily glycosylated. Pse (Pseudaminic Acid Bacterial Glycans) and Pse-like glycans have also been identified as an obligate precursor for glycosylation reactions involved in flagellar production. *PseI* (*TDE0960*) in *T. denticola* is critical for *flaA* and *flaB* expression and flagellin biosynthesis [100]. Reduction of FlaA and FlaB proteins in deletion mutants resulted in impaired assembly of flagellar filaments, ultimately leading to aberrant morphogenesis, lack of motility, reduced adherence and host cell invasion [100].

## 5. Conclusions and Recommendations for Future Research

An enhanced understanding of the mechanisms which drive the transition of periodontal pathobionts from commensals to pathogens is critical for the development of prevention strategies and therapies for periodontitis. This review has identified genetic, transcriptional and post-translational mechanisms that regulate the pathogenic potential of *P. gingivalis*. There is a distinct lack of literature regarding equivalent mechanisms in other periodontal bacteria which are considered equally critical to the pathogenesis of periodontal disease, namely *Treponema denticola*, *Tannerella forsythia, Prevotella intermedia and Aggregatibacter Actinomycetemcomitans.* Oral bacteria interact and synergise to enhance their pathogenicity under less favourable conditions. Furthermore, horizontal gene transfer occurs between different bacterial species that colonise the periodontal environment. Therefore, the scope of identifying modulatory mechanisms of bacterial virulence involved in periodontal disease must be expanded to include other periodontal pathobionts. 

Moving forward, additional research is needed to establish the mechanisms by which periodontal pathobionts alter their virulence and cause disease. On the backdrop of long-standing literature investigating the genetic diversity of *P. gingivalis*, the evidence on the relationship between genetic diversity and altered pathogenicity of *P. gingivalis* is conflicting. On the other hand, the literature concerning transcriptional regulation and post-translational modifications of virulence factors is relatively limited and merits further study.

The field of epigenetics is of significant interest in the study of bacterial pathogenicity. Bacteria such as *Streptococcus ssp*. [113], *Helicobacter pylori* [114] and *Neisseria gonorrhoeae* [115] utilise epigenetic modifications for the modulation of virulence and immune evasion by switching to allelic forms that permits their adaptation to changing environments. In bacterial phase variation, DNA methylation driven by environmental change modulates gene expression and phenotypic diversity without mutations or genetic rearrangements of sequences encoding the virulence factors themselves. Epigenetic phase variations may be equally important in periodontal pathogens, as these mechanisms mediate virulence and immune evasion to aid the survival of a range of bacterial species under environmental stresses [116,117].

## Figures and Tables

**Figure 1 microorganisms-11-00015-f001:**
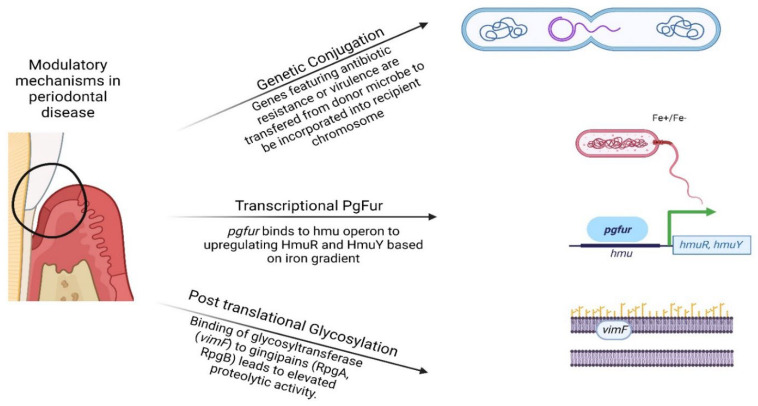
Visual representation of the genetic, transcriptional, and post-translational modifications of periodontal pathobionts presented in this review. (Created by BioRender.com) accessed on 20 October 2022.

**Table 1 microorganisms-11-00015-t001:** Summary of modulatory mechanisms of pathogenicity employed by *P. gingivalis* and other periodontal pathobionts referenced in this review.

Species	Mechanism	Description	References
*P. gingivalis*	Fimbrial Allele variation	Allelic variations of *fimA* (type I, II, III and IV) are implicated in different pathogenic phenotypes.	[31,32,33,34,35]
*P. gingivalis*	*rag* locus variation	*Rag 1–4* are implicated in modulation of virulence.	[36,37,38,39]
*P. gingivalis*	Absence of the *rag* locus Loss of capsule	Loss of *rag* gene or the external capsule leads to loss of virulence and retarded survival.	*rag* [40,41,42,43] Capsule [44,45,46,47]
*P. gingivalis*	Conjugation of Transposons (CTnPg1/ IS)	Incorporation of transposons leads to improved bacterial survival.	[48,49,50,51,52,53,54]
*P. gingivalis*	CRISPR-Cas system	*Cas3* implicated in mediation of bacterial metabolism and immune homeostasis.	[55,56,57,58,59,60]
*P. gingivalis*	Outer Membrane Vesicles (OMVs)	Transfer of virulence genes to other bacteria through extracellular vesicles.	[61,62,63,64,65,66,67,68,69,70]
*P. gingivalis*	PgFur	Ferric uptake regulator, regulates gene expression based on iron gradient. Dysregulation leads to reduced host invasion and adherence.	[71,72,73,74,75]
*P. gingivalis*	PgRsp	Regulator of oxidative stress response, its function depends on iron or haem availability. Regulates expression of virulence genes such as *kpg*.	[76,77,78]
*P. gingivalis*	RprY	Regulates sodium-dependent ubiquinone oxidoreductase system, responsible for clearance of radicals that limit survival in aerobic conditions.	[79]
*P. gingivalis*	ECF	Regulates bacterial response to oxidative stress, through elevated expression of virulence factors	[80,81,82,83,84]
*P. gingivalis*	Protein acetylation	Acetylation of gingipains enhances bacterial survival and virulence.	[85,86,87,88,89,90]
*P. gingivalis* *T. forsythia* *T. denticola*	Protein glycosylation	Acts through Type- IX Secretion System altering virulence, immune response and bacterial metabolism.	[91,92,93,94,95,96,97,98,99,100]

## Data Availability

Not applicable.

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
