# Peer review of "Modulatory Mechanisms of Pathogenicity in Porphyromonas gingivalis and Other Periodontal Pathobionts"

_microorganisms, 2022, doi:10.3390/microorganisms11010015_

Round 1

Reviewer 1 Report

Dear Authors,

The present article summarizes the genetic mechanisms of  P. gingivalis in periodontal disease.  I suggest a modification of the title in order to reflect the content.

Line 35 . This review will present . Use the present time not future.

From lines 120 to 136 there is no reference in the manuscript.

The Conclusions need reduction. In the present form they do not synthesize the manuscript and even mention Campylobacter jejuni.

Best regards!

Reviewer 2 Report

Review Microorganisms-2013266 – Pathobionts

 The manuscript (MS) addresses putative mechanisms of periodontal pathogenicity of Porphyromonas gingivalis. The focus is on the genes, and on experiments with knockout mutants and characterization of variants. The conundrum, that the pathogenesis of periodontal disease is incompletely understood, is candidly expressed (Line 49-50), as are contradictory findings (e.g. L. 141-3, L. 150-2, L. 260-5).

The MS is well written and clearly presented. As almost all data relates to P. gingivalis (L.486-7: “There is a distinct lack of literature regarding equivalent mechanisms in other periodontal bacteria”), the title of the MS could be changed to reflect this fact. Also, Section 5: Conclusions, is a bit ambiguous. After reviewing putative mechanisms, it would be helpful to emphasize and discuss the lack of proof of the possible pathogenesis, rather than to reach out for Helicobacter pylori and Campylobacter jejuni.

One argument is puzzling to this reviewer. In section 2.2 it is mentioned that the presence of short fimbriae could result from horizontal gene transfer from Streptococci. Although the argument is mentioned as a reasonable speculation, reiterative transfers from a completely different phylum is difficult to comprehend, and in contrast with the normal modes of natural transformation.

Specific points.

Bacterial names should be in italics throughout.

Line 87-92. The sentence is a bit confusing when reactive oxygen species – which are oxygen molecules – are mixed with their products (damaged protein and lipids).

L. 93-8. F. nucleatum is emphasized as a decisive bacterial species in biofilm formation, but references are lacking.

Table 1. The Table is complex and takes a lot of space. Perhaps “Pathobiont” could be omitted (only one modification relates to non-gingivalis), and references could be abbreviated (eg [31, 46-51]).

L. 126-7. It is unexpected (to this reviewer) that the author mention DNA in the biofilm matrix in connection with horizontal transfer. Is there any connection between these two circumstances, apart from the presence of DNA molecules?

L. 248. A verb is missing.
